# Identification of Crucial Amino Acids in Begomovirus C4 Proteins Involved in the Modulation of the Severity of Leaf Curling Symptoms

**DOI:** 10.3390/v14030499

**Published:** 2022-02-28

**Authors:** Kao-Wei Dai, Yu-Ting Tsai, Chia-Ying Wu, Yi-Chin Lai, Na-Sheng Lin, Chung-Chi Hu

**Affiliations:** 1Graduate Institute of Biotechnology, National Chung Hsing University, Taichung 40227, Taiwan; jackalgood730711@gmail.com (K.-W.D.); ting10291029@gmail.com (Y.-T.T.); wci823@yahoo.com.tw (C.-Y.W.); yichinlai@gmail.com (Y.-C.L.); 2Institute of Plant and Microbial Biology, Academia Sinica, Taipei 115024, Taiwan; 3Advanced Plant Biotechnology Center, National Chung Hsing University, Taichung 40227, Taiwan

**Keywords:** *Begomovirus*, geminiviruses, leaf curl, symptom determinant, C4 protein, pathogenesis, disease management, symptom regulation

## Abstract

Begomoviruses frequently inflict upward or downward leaf curling symptoms on infected plants, leading to severe economic damages. Knowledge of the underlying mechanism controlling the leaf curling severity may facilitate the development of alternative disease management strategies. In this study, through genomic recombination between *Ageratum yellow vein virus* Nan-Tou strain (AYVV-NT) and *Tomato leaf curl virus* Tai-Chung Strain (TLCV-TC), which caused upward and downward leaf curling on *Nicotiana benthamiana*, respectively, it was found that the coding region of C4 protein might be involved in the determination of leaf curling directions. Sequence comparison and mutational analysis revealed that the cysteine and glycine at position 8 and 14 of AYVV-TC C4 protein, respectively, are involved in the modulation of leaf curling symptoms. Cross-protection assays further demonstrated that *N. benthamiana* inoculated with AYVV-carrying mutations of the aforementioned amino acids exhibited attenuated leaf curling symptoms under the challenge of wild-type AYVV-NT. Together, these findings revealed a new function of begomovirus C4 proteins involved in the modulation of leaf curling severity during symptom formation and suggested potential applications for managing viral diseases through manipulating the symptoms.

## 1. Introduction

The family *Geminiviridae* consists of viruses with single-stranded circular DNA genomes encapsidated in twinned incomplete icosahedral virions [1]. Geminiviruses in the genus *Begomovirus* are transmitted by whiteflies (*Bemisia tabaci* cryptic species complex) and infects dicotyledonous plants, causing severe damages to crops worldwide [2,3,4,5]. Typical symptoms caused by begomoviruses include leaf curling, vein or leaf yellowing, and mosaic [4]. However, the mechanisms underlying the elicitation of specific symptoms and the modulation of disease severity remain largely elusive. Knowledge on the regulation of the formation of specific symptoms may provide deeper insights into the developmental and physiological processes of host plants and pave the way for the development of alternative viral disease management measures through the manipulation of symptom formation.

Several proteins encoded by geminiviruses have been demonstrated to be determinants of specific symptoms. Geminiviruses harbor either one or two genomic DNA components, designated mono- or bi-partite genomes, respectively. The single-stranded circular genomic DNA is ambisense, encoding proteins on both the sense of the virion-encapsidated DNA (V-sense) and on the complementary sense (C-sense). On the V-sense are the open reading frames (ORFs) for coat protein (CP, or V1) and movement protein (MP, or V2). On the C-sense reside the overlapping ORFs for replication-associated protein (Rep, or C1), transcription activator protein (TrAP, or C2), replication enhancer (REn, or C3), and C4, which is completely embedded in the ORF for Rep but on a different frame [1]. C4 protein has been studied extensively in recent years and demonstrated to be an important symptom determinant in geminivirus infections [6,7,8,9,10,11,12,13,14,15,16,17]. In addition to C4 protein, Rep [9,11,18], C2 [19,20,21], and V2 [9,11,15] have also been shown to be symptom determinants in the infection cycles of different geminiviruses. Furthermore, a protein, βC1, encoded by a betasatellite associated with *Tomato yellow leaf curl China virus*, has been demonstrated to interfere with leaf development by directly interacting with the host protein asymmetric leaf 1 (AS1) [22]. However, the determinant(s) for the specific directions and severity of leaf curling symptoms remained to be explored.

Among these known symptom determinants, C4 protein has attracted much attention in recent years. Being the smallest and one of the least-conserved proteins encoded by geminiviruses [8], C4 proteins may exhibit the highest number of functions in infection cycles and pathogenesis, with new functions continuing to be identified [23,24]. The mechanisms of certain functions of C4 protein have been elucidated. For example, C4 proteins may induce the abnormal development of plants by regulating the brassinosteroid signaling pathway through interactions with members of SHAGGY-like protein kinase [7,12,24,25,26]. C4 proteins of certain begomoviruses have been shown to be a viral suppressor of RNA silencing [27,28,29,30]. The suppression was mediated either by direct binding to the single-stranded microRNAs (miRNAs) and small interfering RNAs (siRNAs) [27,31] to block the cleavage or by interacting with plasma membrane proteins BAM1 or BAM2 to interfere with the intercellular spread of miRNAs or siRNAs [8,32,33]. C4 protein has also been reported to be involved in nucleocytoplasmic shuttling [34] and drought resistance [35]. However, it should be noted that the C4 proteins of different geminiviruses do not share equal functions [36], as C4 gene is the least conserved among geminivirus-encoded genes [8]. Despite the richness of the previous studies on C4 proteins of geminiviruses, the role of C4 protein in the determination of leaf curling severity has not been revealed in detail.

Previously, we have developed a simplified method for the generation of infectious constructs of begomoviruses through rolling circle amplification (RCA) and demonstrated the applicability of the method by constructing infectious clones of two mono-partite begomoviruses, *Ageratum yellow vein virus* Nan-Tou strain (AYVV-NT), *Tomato leaf curl virus* Tai-Chung strain (TLCV-TC), designated pBinAYVV, pBinTLCV, respectively, and a bi-partite begomovirus, *Squash leaf curl virus* Yun-Lin strain (SqLCV-YL), designated pBinSqLA and pBinSqLB [37]. The symptoms associated with these begomoviruses on their orginal hosts are yellow veins without leaf curling for AYVV on ageratum, upward leaf curling for TLCV on tomato, and downward leaf curling for SqLCV on muskmelon plants [37]. With the diverse types of leaf curling symptoms, these constructs provide a practical system for studying the mechanisms underlying the determination of leaf curling symptoms with specific directions and severity. For comparison within the same host genetic background, *N. benthamiana* was chosen as the test plant for these constructs in the following inoculation assays.

In this study, we have focused on the analyses of viral determinant(s) for the specific direction or the degree of severity of leaf curling symptoms elicited by specific begomoviruses. By using AYVV-NT, TLCV-TC, and SqLCV-YL as the experimental system, we have mapped the region on the begomovirus genome for the determination of leaf curling directions in *N. benthamiana* and further identified the crucial amino acids in C4 protein involved in the regulation of the severity of leaf curling symptoms. The applicability of mutants carrying alterations at these two amino acids in the modulation of leaf curling symptoms was also assayed. Taken together, these studies provided further insight into the regulation of leaf curl symptom formation and suggested potential usages in viral disease management.

## 2. Materials and Methods

### 2.1. Plant Materials and Virus Inoculation

The aforementioned infectious clones of three begomoviruses, AYVV-NT, TLCV-TC, and SqLCV-YL [37] were used as the starting material for the construction of other chimeric viruses or mutants to investigate the viral factors involved in the determination of the severity of leaf curling symptoms. *N. benthamiana* plants, at the stage with 6–7 fully expanded leaves, were used as the assay hosts. Plants were cultivated in a growth room maintained at 25–28 °C, with 16 h light/8 h dark cycles. The infectious constructs of the wild-type viruses or mutants were inoculated into *N. benthamiana* plants by using *Agrobacterium*-mediated infection as described previously [37,38,39,40]. *A. tumefaciens* cells harboring various constructs were adjusted to an optical density of 1.0 at 600 nm (OD_600_ = 1.0) and infiltrated into the back side of the leaves using 1-mL syringes. Symptoms were recorded at 14, 21, and 28 days post inoculation (dpi). The nucleotide sequences of the viral progenies were further verified by DNA extraction from upper un-inoculated leaves and sequencing as described in the following sections.

### 2.2. Plasmid Construction and RCA

The plasmids used in this study were constructed essentially following the protocols as described previously [37]. Mutant constructs were generated by inverse-polymerase chain reaction (inverse-PCR) [41] using specific primer pairs to introduce the mutations as specified in the relevant section using the circular genomic DNA of the wild-type viruses as the templates. The inverse-PCR makes use of the circular template and a pair of back-to-back primers pointing to the inversed directions as opposed to those used in the conventional PCR. The primers were designed with the mutations and could be used to generate linear full-length DNA of the circular template by PCR. The amplified products containing the desired mutations were then self-ligated into a circular form, digested with unique restriction enzyme (*Bam*HI in our case), and inserted into the cognate restriction enzyme sites in vector pUC119 [42]. The full-length genome of the virus harboring the desired mutations were then released from pUC119 [42] by restriction enzyme digestion, self-ligated into circular form, and subjected to RCA as described previously [37,43]. The amplified products were partially digested with restriction enzyme *Bam*HI to generate dimers of the viral genome, since at least two copies of the origin of replication were required for the construct to be infectious, and subsequently cloned into the *Bam*HI site of the vector pBin19 as described previously [37], followed by nucleotide sequencing to confirm the presence of the desired mutation(s). To confirm the infectivity and the maintenance of mutations in the progenies of the infectious constructs, total DNA was extracted from the newly emerged leaves of the inoculated *N. benthamiana*, and subjected to RCA as described previously [37,43]. The amplified DNA was digested with restriction enzyme *Bam*HI, cloned into pUC119 vector [42], and subsequently sequenced. At least 10 independent clones of the progeny viruses were selected for sequencing analysis to verify the nucleotide sequences.

### 2.3. Sequence Analyses

The full-length sequences of AYVV-NT, TLCV-TC, and SqLCV-YL genomes have been deposited in GenBank under the accession numbers EF458639, MZ713252, and EU479710/EU479711 (for SqLCV-YL DNA A/B) respectively. Multiple sequence alignment of different C4 protein amino acid sequences encoded by the above begomoviruses was performed using the software CLUSTAL W [44] with the default settings (using the Gonnet scoring matrix with a gap opening penalty of 10 for both pairwise and multiple alignments and gap extension penalties of 0.1 and 0.2 for pairwise and multiple alignments, respectively) and the alignments were presented using the software GeneDoc [45]. For the design of primers for creating mutations in C4 proteins without affecting the amino acid sequences of the overlapping Rep protein on another reading fame, the software BioEdit [46] was used.

## 3. Results

### 3.1. Identification of the Determinant of Leaf Curling Direction by Genome Recombination

As an initial step to analyze the viral factors involved in the determination of specific leaf curling symptoms, we examined the ability of a specific begomovirus to inflict leaf curling symptoms with specific direction on the same host, *N. benthamiana,* to standardize the host background. *A. tumefaciens* cells harboring the infectious constructs of AYVV-NT, TLCV-TC, and SqLCV-YL [37] were infiltrated into *N. benthamiana*. As shown in Figure 1, pBinAYVV induced severe upward leaf curling symptoms, whereas pBinTLCV and pBinSqLA + pBinSqLB elicited downward leaf curling symptoms on the young leaves of the *N. benthamiana* plants inoculated. Although the symptoms on *N. benthamiana* inflicted by these infectious constructs were different from those on their original host plants, the directions of leaf curling symptoms remained consistent. Thus, these infectious clones provided useful starting materials for the analyses of the determinant of leaf curling direction.

To identify the genomic region involved in the determination of the direction of leaf curling symptoms, we adopted the strategy of genomic recombination. The prerequisite for creating chimeric geminiviruses is that the iteron sequences, required for the recognition by Rep protein to initiate replication, of the two viruses to be recombined should be the same or highly similar, so that the Rep proteins of the parental viruses may still recognize those of the recombinants. Therefore, we performed sequence analysis and found that AYVV-NT and TLCV-TC share identical iteron sequences, 5′-GGTGTCTC-3′ and 5′-GGTGTACT-3′, for recognition by Rep protein for specific replication [47,48], suggesting that these two viruses are suitable candidates for genome recombination. Since C4 proteins have been implicated as pathogenesis determinants in recent studies [7,8,12,17], our initial attempt was to exchange the C4 protein coding regions between AYVV-NT and TLCV-TC. Thus, chimeric viruses between AYVV-NT and TLCV-TC were constructed as illustrated in Figure 2. The unique restriction sites for NcoI and BamHI were used for the exchange of genome fragments (the inserts) from positions 1969 and 136, respectively. The genomic DNA of AYVV-NT or TLCV-TC was amplified by RCA and digested with NcoI and BamHI to generate the respective backbones and inserts. The inserts were then exchanged and ligated to different backbones and cloned as monomers into the vector pUC119 [42]. The recombinant virus with TLCV-TC backbone harboring AYVV-NT insert was designated TA, and the one with AYVV-NT backbone and TLCV-TC insert was designated as AT. Two independent clones were selected for each chimeric virus (TA3-4, TA3-12 for TA chimera and AT5-20 and AT5-24 for AT chimera). Following the verification of the recombination through nucleotide sequencing, the monomeric genomes were released from pUC119 by digestion with BamHI, self-ligated into a circular form, and amplified by RCA. Following partial digestion of the RCA products with BamHI, the dimeric genomic fragments were cloned into the vector pBin19 as described previously [37] to generate the constructs, pBinTA3-4, pBinTA3-12, pBinAT5-20, and pBinAT5-24, used for inoculation assays. These constructs were inoculated into *N. benthamiana* plants through *Agrobacterium*-mediated infiltration, and the progenies of the chimeric viruses were sequenced to verify the maintenance of the recombination. The results of the inoculation assay are summarized in Table 1. Representative symptoms caused by these constructs are shown in Figure 3. For unknown reason, pBinTA3-4 was not infectious, possibly due to problems that happened during the construction of dimeric constructs, and the infectivity of pBinAT5-20 was slightly lower than that of pBinAT5-24 (Table 1). Nevertheless, the results clearly showed that the direction of leaf curling symptom was determined by the genome fragment from nucleotide positions 1969 to 136 (highlighted by the green-lined boxes in Figure 2). All plants successfully infected with viruses harboring the AYVV-NT inserts exhibited severe upward leaf curling symptoms, whereas those infected by viruses with TLCV-TC inserts developed downward leaf curling symptoms (Table 1). This insert region encompass the full open reading frame (ORF) of C4 protein, the partial ORF for the N-terminal half of Rep protein, and the intergenic region (IR) including the origin of replication (indicated by the stem-loop in Figure 2). Thus, further experiments were performed to identify the major determinant of leaf curling direction.

We have used several approaches to directly test the function of C4 proteins of AYVV-NT and TLCV-TC in the modulation of leaf curling directions by expressing different C4 proteins in plant, either transiently [49,50] or through transgenic plants. However, these approaches have been unsuccessful. Thus, to distinguish the effect of C4 proteins from the Rep protein or IR sequences on the modulation of leaf curling directions, we performed a loss-of-function experiment. The start codon of the C4 protein ORFs of both chimeric viruses were mutated to a stop codon to disrupt the expression of C4 proteins without affecting the translation of the overlapping ORF for Rep protein by using primers AT3-AC4mut F (5′-CTCGCTAGCTCGTGCAATTCTCTGCAGAT-3′, with the mutated nucleotide underlined), AT3-AC4mut R (5′-CGAGCTAGCGAGCCCTCATCTCCACGTGC-3′), TA5-AC4mut F (5′-GTCGCTAGCTCGTGAAGCTCTCTGCAAAC-3′), and TA5-AC4mut R (5′-CGAGCTAGCGACTCCTCACCTGCACATTC-3′) through inverse PCR [41], followed by RCA, and cloned into pBin19 as described above. However, the mutation abolished the infectivity of these constructs, confirming that the expression of C4 proteins is important for the viruses to complete their infection cycles. It has been reported that the premature termination of the C4 protein expression of the *Beet curly top virus*, a geminivirus in the genus *Curtovirus*, may cause the change of directions of leaf curling symptoms from upward to downward [16], indicating that the C4 protein of curtoviruses is involved in the modulation of leaf curling directions. The involvement of the C4 protein of begomoviruses in the control of leaf curling tendencies has not been reported previously. Thus, we performed the following point mutational analyses to corroborate the hypothesis that the C4 proteins of AYVV-NT and TLCV-TC are involved in the determination of leaf curling directions.

### 3.2. In Search of Amino Acids Involved in the Modulation of Leaf Curling Symptoms

To fine-map the critical amino acids for the development of leaf curling symptoms with different directions, the C4 protein amino acid sequence of AYVV-NT (GenBank Accession number EF458639), which causes an upward leaf curling symptom, and those of TLCV-TC and SqLCV-YL, which elicit a downward leaf curling symptom [37], were aligned using CLUSTAL W [44]. As shown in Figure 4, candidate positions were highlighted in which the amino acids in AYVV-NT differed from that conserved in TLCV-TC and SqLCV-YL. These highlighted amino acids were selected based on the criterion that the codons for these amino acid should allow for mutations that change the ones in C4 protein of AYVV-NT into those in the corresponding positions in C4 proteins of TLCV-TC and SqLCV-YL without affecting the amino acids of Rep protein encoded on another reading frame. To test the functions of the amino acids in these positions on the determination of the leaf curling directions, specific mutations were generated in the C4 protein of AYVV-NT infectious clone pBinAYVV for inoculation assays. These positions were divided into three groups for mutational study to simplify the experimental design. In A-mut of AYVV-NT, the cysteine and glycine at positions 8 and 14 were replaced with phenylalanine and glutamic acid, respectively, conserved for TLCV-TC and SqLCV-YL (abbreviated as C8F and G14E). Similarly, B-mut contains the phenylalanine to serine exchange at position 64 (F64S). C-mut harbors glutamine to leucine, aspartic acid to valine, and methionine to threonine mutations at positions 71, 74 and 79 (Q71L, D74V, and M79T), respectively (Figure 4). Since the open-reading-frame of C4 protein overlaps with that for Rep protein, but on a different frame, the mutations in AYVV C4 protein were designed to maintain the original amino acids in Rep protein, which is required for the initiation of geminivirus replication. Thus, for the generation of A-mut, the primer pair, AY-A-mut-F, 5′-CTGGATAAGAACGTGGAGATGA-3′, AY-A-mut-R, 5′-TTCGAAGGAAAATACCAGTGCA-3′, were used to introduce the C8F and G14E mutations in the C4 protein, while maintaining the V (GTG to GTT) and G (GGG to GGA) in the Rep protein. By the same token, the primer pair, AY-B-mut-F, 5′-CCCATTCGAGGGTGTCTC-3′ and AY-B-mut-R, 5′-GTGAGTTCCAGATCGATG-3′ were used to generate B-mut; while primer pair AY-C-mut-F, 5′-TGTTGACCTCCTCTAGCAG-3′ and AY-C-mut-R, 5′-GACAGCCAACGACGCTTAC-3′ were used for creating C-mut. The mutants were constructed by inverse-PCR using the aforementioned primer pairs with the genomic DNA of AYVV-NT as the template [37], verified by nucleotide sequencing. The mutants of AYVV-NT were subsequently inoculated into *N. benthamiana* through *Agrobacterium*-mediated infection to examine the effects on leaf curling symptom. The results of inoculation assays are summarized in Table 2, with representative phenotypes shown in Figure 5. It was found that *N. benthamiana* plants inoculated with A-mut of AYVV-NT showed mild symptoms, with the margins of the young leaves slightly curled up (Figure 5) at 24 dpi. In contrast, B-mut and C-mut of AYVV-NT elicited severe upward leaf curling symptoms, similar to those caused by the infectious clone of wild-type AYVV-NT, pBinAYVV (Figure 5). The plants inoculated with the empty vector pBin19 or buffer did not show any symptom, as expected. To verify the maintenance of mutations, the genomic DNA of the progenies of A-mut, B-mut, and C-mut was extracted from the upper un-inoculated leaves, amplified by RCA [43], and fully sequenced. It was confirmed that the mutations were maintained in the progenies, supporting the notion that the symptoms indeed resulted from the infection of the mutants. The results indicated that the hypothesis that these three groups of amino acids in C4 proteins are involved in the determination of leaf curling directions could not be confirmed. Nevertheless, these observations demonstrated that Cys8 and/or Gly14 play a role in the severity of upward leaf curling. In contrast, Phe64, Gln71, Asp74, and/or Met79 may not contribute significantly to the severity of leaf curling symptom in the current experimental settings, as shown by the results of inoculation assays with B-mut and C-mut.

### 3.3. Potential Application of A-Mut in Disease Management

The above results demonstrated that A-mut could effectively attenuate the upward leaf curling symptoms caused by the original pBinAYVV construct. To test whether A-mut could be utilized as preventive or therapeutic agents for the treatment of severe symptoms caused by pBinAYVV infections, inoculation assays were performed with different inoculation orders for pBinAYVV and A-mut. For the prevention of the formation of severe upward leaf curling symptoms, *A. tumefaciens* cells harboring A-mut construct were first infiltrated into *N. benthamiana* for 7 days, followed by the challenge of pBinAYVV. For use of A-mut as the therapeutic agent, *A. tumefaciens* cells harboring pBinAYVV were first infiltrated into *N. benthamiana*, followed by the inoculation of A-mut at 7 dpi. The result of the inoculation assay is summarized in Table 3, with the representative symptoms caused by different treatments shown in Figure 6 and Appendix A. The result revealed that A-mut could indeed be used as a preventive treatment to avoid the formation of severe upward leaf curling symptoms caused by pBinAYVV. The preventive effect was maintained for at least 34 days (Figure 6 and Appendix A). However, A-mut could not be used as a therapeutic to heal or attenuate the severe upward leaf curling symptoms caused by pBinAYVV. To examine the progeny population ratio of A-mut and pBinAYVV in the “prevention” and “therapeutic” treatments, total DNA was extracted from the upper un-inoculated leaves, amplified by RCA, and cloned, for Experiment I and III. From each treatment, 10 independent clones were sequenced. It was found that 18 out of a total of 20 clones from the two independent experiments in the “prevention” treatment were A-mut, whereas all 20 clones were pBinAYVV in the “therapeutics” treatment. It should be noted that, in the “prevention” treatments, one out of four plants in each experiment was not “protected” and exhibited severe upward leaf curling symptoms following the challenge of pBinAYVV (the representative plant indicated by the white arrow in Appendix A). Sequencing analysis of 10 clones from the progeny viruses in Experiment III revealed that only AYVV-NT was present in this plant. This observation provided evidence suggesting that the “preventive effect” of A-mut was dependent on the successful establishment of A-mut prior to the challenge of pBinAYVV and not from the wound-induced resistance through the inoculation process. The results reflected the lower infectivity observed for A-mut (Table 2) compared to pBinAYVV and suggested that the ability of the C4 protein to modulate the symptoms is tightly associated with the accumulation levels of the respective virus and the C4 protein.

## 4. Discussion

The C4 proteins encoded by geminiviruses are multifunctional, with ever-expanding roles being identified continuously [23]. However, the functions of C4 proteins are not conserved for all geminiviruses [36]. In this study, we have revealed an additional function of C4 proteins in the modulation of the severity of the leaf curling symptoms of AYVV-NT and TLCV-TC. Two amino acids, Cys8 and Gly14, of AYVV-NT C4 protein were further fine-mapped as playing crucial roles in modulating the severity of upward leaf curling symptoms caused by AYVV-NT. We further tested the possibility of using A-mut as a preventive or therapeutic measure against the infection of wild-type AYVV-NT. The findings in this study may contribute to the understanding of the mechanisms of symptom formation, which may further be applied in the development of alternative methods to manage viral diseases.

One of the initial goals of this study was to identify the viral determinant(s) of the specific directions of leaf curling symptoms caused by the infection of specific begomoviruses. We have created chimeric viruses by exchanging partial genomic regions between AYVV-NT and TLCV-TC and found that the region between the unique *Nco*I and *Bam*HI restriction sites (Figure 2) exhibited the ability to modulate leaf curling directions in *N. benthamiana* (Figure 3). We then focused on the only intact open reading frame, C4, in this region for the following mutational analyses, assuming that the C4 protein might be the viral component for the determination of specific leaf curling direction. However, this assumption might be premature, as the exchanged genomic region also contains partial coding sequence for the Rep protein and the non-coding sequences (Appendix A) required for replication initiation (*Ori*, iterons) and gene expression (promoters). The possibilities that these viral factors, other than C4 protein, may be involved in the determination of leaf curling directions could not be ruled out. It is also possible that the other viral proteins may also be required for specific symptom development, since it has been shown that the expression of C4 protein alone may not necessarily recreate the symptoms caused by the specific virus, and C4 proteins encoded by different begomoviruses may exhibit different abilities in symptom elicitation [7]. Further experiments are required to resolve this issue.

In the assays with A-mut as the “preventive“ or “therapeutic” measure in mitigating the severe upward leaf curling symptoms caused by wild-type AYVV-NT in *N. benthamiana*, the initial results seemed to indicate that the mutations in Cys8 or Gly14 in AYVV-NT C4 protein may only be used as an preventive measure, since the competitiveness of the A-mut was lower compared to that of wild-type AYVV-NT (Table 2 and the sequencing results of the progenies in the “therapeutic” experiments). However, the possibility of using A-mut as a therapeutic agent has not been ruled out, as the conditions for the application of A-mut have not been exhaustively tested. For example, inoculation of *A. tumefaciens* cells harboring A-mut with a much higher concentration may increase the competitiveness of A-mut, which might ultimately exhibit the therapeutic effect. In addition, several controls should be included in these assays, such as pre-inoculation with pBin19 vector alone followed by inoculation with AYVV-NT or by using test plants in different developmental stages. These controls are important to test thoroughly the abilities of A-mut in the modulation of the severity of leaf curling symptoms. These experiments will be performed in our forthcoming studies. On the other hand, the reciprocal A-, B-, and C-mutations in TLCV-TC were unable to be designed and tested, since these mutations in C4 protein of TLCV-TC would also affect the amino acids of Rep protein in the overlapping open reading frame. Thus the effect of amino acids in these positions on the modulation of symptoms could not be thoroughly tested. Nevertheless, the current results indicated that amino acids Cys8 and Gly14 in the AYVV C4 protein are involved in the modulation of the severity of the upward leaf curling symptoms caused by AYVV-NT in *N. benthamiana*. Additional experiments are required to comprehend the mechanism of leaf curling symptom formation and develop alternative measures to manage the leaf curling disease.

To further explore the possible function of Cys8 and Gly14 in AYVV-NT C4 protein, we have performed 2- and 3-dimensional (2D and 3D) structural predictions using Phyre2 web portal (http://www.sbg.bio.ic.ac.uk/phyre2/html/page.cgi?id=index, as accessed on 20 January 2022) [51]. For 3D prediction, no suitable templates were identified for modeling the structures in this region. However, for 2D analysis, Cys8 is predicted to be located at the first beta-strand and Gly14 in the coil region, similar to those for the corresponding amino acids in TLCV-TC and SqLCV-YL (Appendix A). The mutations in A-mut did not affect the predicted secondary structure (Appendix A), suggesting that the difference in leaf curling severity might not result from the difference in C4 protein structures. Instead, amino acids in this region might be involved in the interaction with specific host factors. Possible candidates of such host factors involved in the regulation of directions for leaf curling symptoms include host proteins and miRNAs that have been reported to modulate the differentiation and development process of leaf tissues [12,17,22,23,24,25,30,52,53,54,55,56,57,58]. Whether the C4 proteins of AYVV-NT or TLCV-TC exhibit differential interactions with these host factors will be one of the focuses in our upcoming studies.

To test the ability of C4 proteins of AYVV-NT and TLCV-TC in the modulation of leaf curling directions independent of other viral factors, expressing the C4 proteins in plants is the direct approach for confirmation. Several attempts have been made in our earlier studies to express the C4 protein of AYVV and TLCV in *N. benthamiana*, either transiently using *Bamboo mosaic virus* (BaMV)- or satellite BaMV RNA-based expression vectors [49,50] or by transgenic lines. However, these attempts have been unsuccessful, possibly because the levels of C4 protein overexpression posed certain toxic effects on the plant cells. Plants inoculated with the over-expressing constructs usually died or exhibited severe leaf necrosis symptoms before the development of any leaf curling symptoms. Thus, we mutated the strat codon of the C4 gene to stop codon in the infectious construct to analyze the functions of the C4 protein in manipulating leaf curling symptoms. However, the mutant viruses without the expression of C4 proteins were not infectious. Therefore, other approaches in directly testing the functions of C4 proteins, such as the infiltration of leaves using a much smaller amount of *A. tumefaciens* cells harboring the C4 protein expression constructs or with other expression systems will be applied in our upcoming studies.

It is worth noting that the C4 proteins of different begomoviruses may have different subcellular localization, which may significantly affect the functionality of different C4 proteins during infection [36]. Therefore, the difference in the subcellular localization of the C4 proteins of different begomoviruses inflicting leaf curl symptoms in specific directions should also be verified. Together, knowledge on the regulation of leaf curling directions or severity by different begomovirus C4 proteins may contribute to our understanding of plant developmental processes and the development of management measures for virus diseases.

## 5. Conclusions

In conclusion, a new function of begomovirus C4 proteins in affecting the severity of leaf curling symptoms has been elucidated in this study. In addition to the previously known functions in viral infection cycles, begomovirus C4 proteins may serve as the modulator of the severity of leaf curling symptoms. The amino acids Cys8 and Gly14 of AYVV-NT C4 protein were found to be involved in the development of leaf curling symptoms elicited by AYVV-NT in *N. benthamiana*. The application of A-mut in “preventing” the development of severe upward leaf curling symptoms caused by AYVV-NT was also demonstrated. Taken together, the results of this study opened up an avenue for further dissecting the processes of leaf curling symptom elicitation and developing new strategies for the management of begomovirus infections.

## Figures and Tables

**Figure 1 viruses-14-00499-f001:**
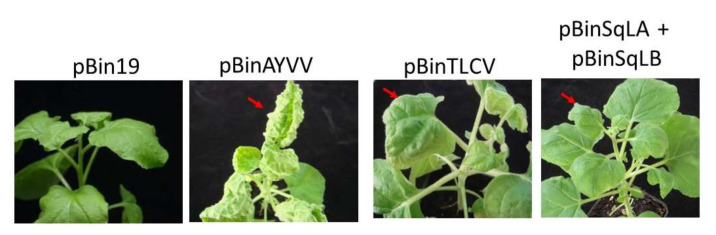
Symptoms on *N. benthamiana* caused by infectious constructs of specific begomovirus. *A. tumefaciens* cells harboring the infectious constructs of various begomoviruses as indicated on the top were infiltrated into the leaves of *N. benthamiana*. The symptoms were recorded at 21 days post inoculation (dpi). The prominent leaf curling symptoms were indicated by the red arrows. pBin19, vector only control; pBinAYVV and pBinTLCV, infectious constructs of *Ageratum yellow vein virus* strain NT (AYVV-NT) and *Tomato leaf curl virus* strain TC (TLCV-TC), respectively; pBinSqLA + pBinSqLB, infectious constructs of DNA A and B, respectively, of *Squash leaf curl virus* strain YL (SqLCV-YL).

**Figure 2 viruses-14-00499-f002:**
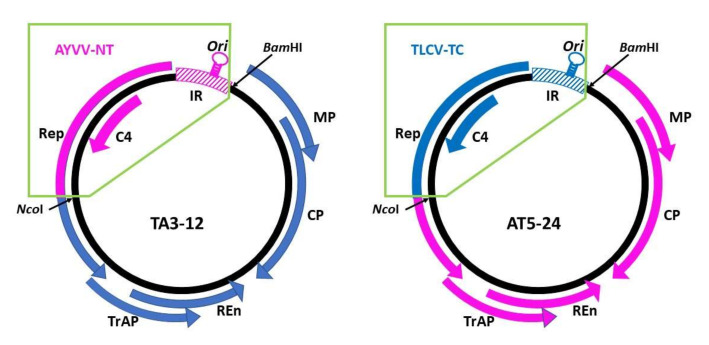
Schematic representation of the construction of chimeric begomovirus between AYVV-NT and TLCV-TC. The circular genomes of the representative chimeric begomoviruses, TA3-12, and AT5-24 were shown, with the genome fragment from AYVV-NT in pink and that from TLCV-TC in blue. The six open reading frames on viral genome were shown as the arrows, with the direction of translation represented by the arrow-head. The identities of the open reading frames were indicated. The restriction enzyme recognition sites, *Bam*HI and *Nco*I, used for the construction of the chimeric virus were specified. The intergenic region, IR, and the origin of replication (*Ori*) were represented by the hashed box and the stem-loop, respectively.

**Figure 3 viruses-14-00499-f003:**
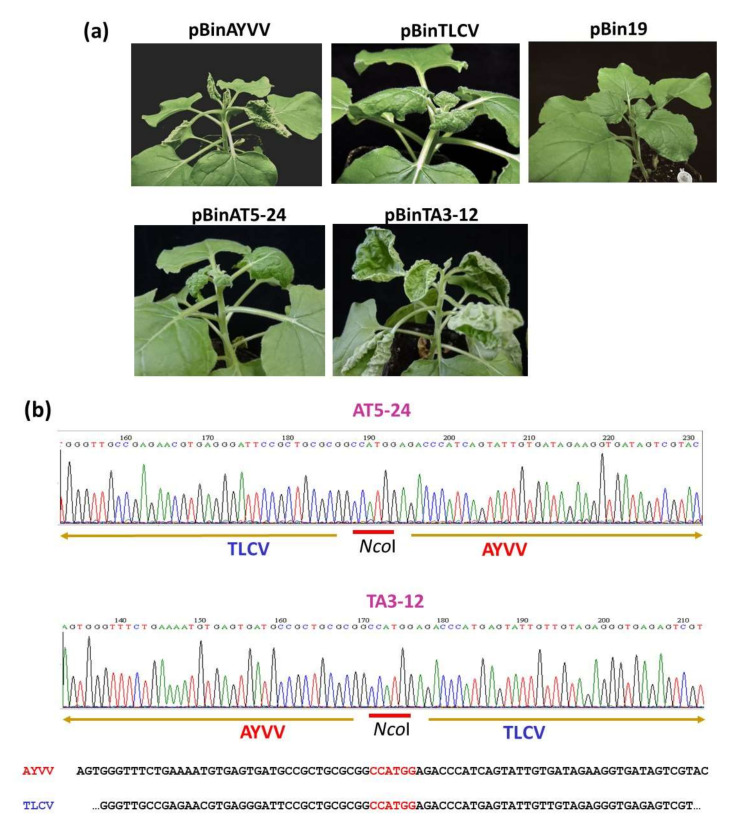
Symptoms induced by various infectious clones of wild-type or chimeric begomoviruses on *Nicotiana benthamiana*. (**a**) *Agrobacterium tumefaciens* cells harboring infectious clones of various begomoviruses were infiltrated into *N. benthamiana*. The symptoms were recorded at 21 days post inoculation (dpi). (**b**) Representative sequencing chromatograms for the analysis of the progenies of the recombinants. To verify the maintenance of the recombination in the progenies, total DNA was extracted from the upper leaves, amplified by RCA, cloned, and sequenced. The nucleotide sequences flanking the NcoI recombination site (indicated by the red bar) were presented. pBin19: empty vector control; pBinAYVV: AYVV-NT wild-type; pBinTLCV: TLCV-TC wild type; pBinAT5-24: chimeric AYVV carrying TLCV C4 gene; pBinTA3-12: chimeric TLCV carrying AYVV C4 gene. The nucleotide sequneces originating from AYVV-NT or TLCV-TC were indicated by the arrows underneath. The original nucleotide sequences of AYVV-NT and TLCV-TC corresponding to those shown in the sequencing chromatograms were shown at the bottom for comparison.

**Figure 4 viruses-14-00499-f004:**
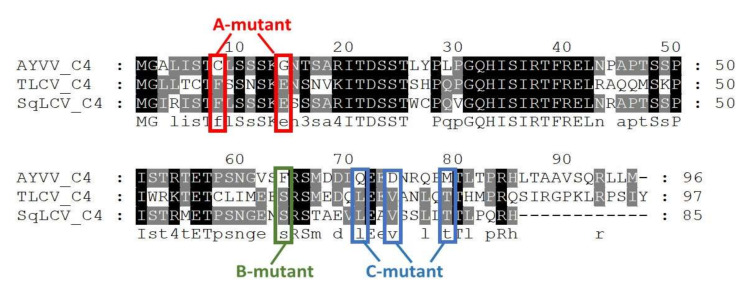
Alignment of C4 protein amino acid sequences of specific begomovirses. The amino acid sequences of C4 proteins encoded by begomoviruses inflicting leaf curl symptoms in opposite directions on *N. benthamiana* were aligned using the CLUSTAL W program [44] and represented using GeneDoc [45]. Among the begomoviruses shown, AYVV-NT caused severe upward leaf curling, whereas TLCV-TC and SqLCV-YL induced downward leaf curling. The amino acids selected to create mutants in subsequent analyses were boxed and indicated. The consensus sequence is shown at the bottom of the alignment. The degrees of conservation were indicated by the background color: black, 100% conservation; gray, conserved in two sequences; white, non-conserved.

**Figure 5 viruses-14-00499-f005:**
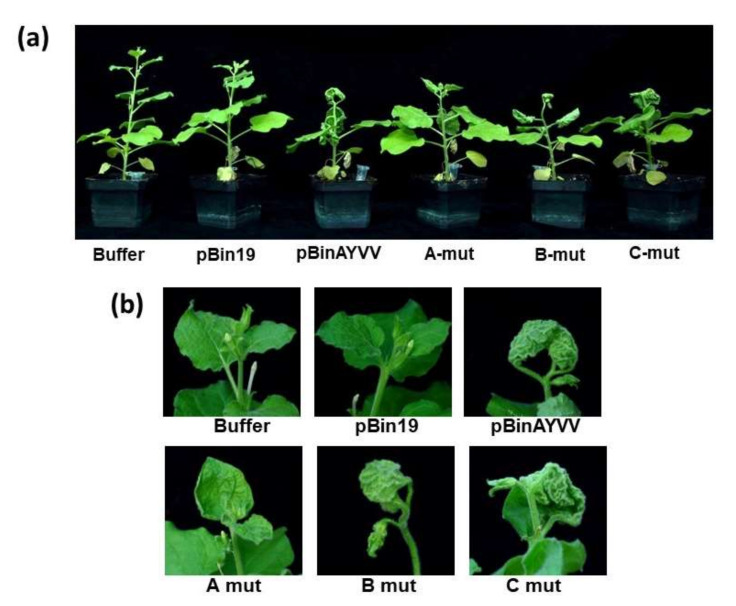
Inoculation assay of AYVV-NT mutants on *N. benthamiana*. *A. tumefaciens* cells (OD_600_ = 1.2) harboring various infectious constructs of AYVV-NT mutants as indicated at the bottom were inoculated into *N. benthamiana*. (**a**) Side view of the whole plants infiltrated with different construct. (**b**) Close-up view of the young leaves of the infiltrated plants. The symptoms were recorded at 24 dpi.

**Figure 6 viruses-14-00499-f006:**
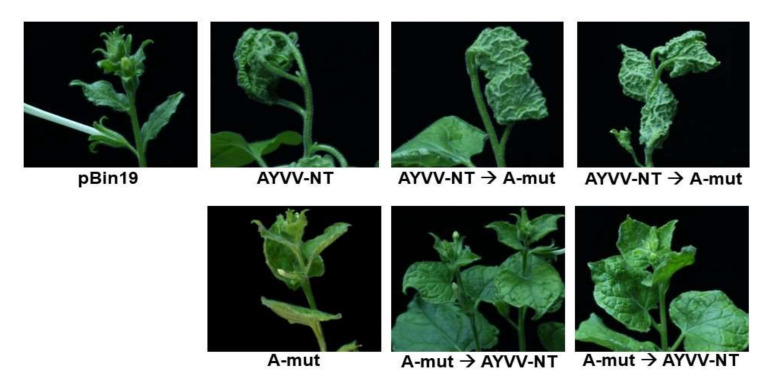
Inoculation assay on the ability of A-mut in the modulation of leaf curling symptom. To test whether A-mut could be used as a potential “preventive” or “therapeutic” agent for begomovirus-induced leaf curling symptoms, *N. benthamiana* plants were first inoculated with AYVV-NT or A-mut, followed by the infiltration of either A-mut or AYVV-NT, respectively, as indicated at the bottom, at 7 dpi, with the arrows representing the order of the infiltration. The effects on leaf curling symptoms were recorded at 34 dpi.

**Table 1 viruses-14-00499-t001:** Inoculation assay of various infectious constructs of wild-type and chimeric viruses of AYVV and TLCV in *N. benthamiana* through *Agrobacterium*-mediated infiltration.

		Inoculation Efficiency ^2^						
		Experiment I			Experiment II			
Inoculum	Leaf Curling ^1^	14 dpi	21 dpi	28 dpi	14 dpi	21 dpi	28 dpi	Infectivity ^3^
pBin19	-	0/4	0/4	0/4	0/10	0/10	0/10	0
pBinAYVV	Up	2/2	2/2	2/2	10/10	10/10	10/10	100
pBinTLCV	Dw	2/2	2/2	2/2	10/10	10/10	10/10	100
pBinTA3-12	Up	0/3	2/3	2/3	0/8	4/8	7/8	82
pBinAT5-20	Dw	2/3	2/3	2/3	6/8	6/8	6/8	73
pBinAT5-24	Up	2/3	2/3	2/3	78	7/8	7/8	82

^1^ Direction of leaf curling symptom: Up, upward; Dw, downward. ^2^. Inoculation efficiency is expressed as the number of plants showing leaf curling symptoms/total number of plants inoculated. Symptoms were recorded at different days post inoculation (dpi) as indicated on the top. Two independent experiments were performed for each inoculum. ^3^ Percentage of total plants showing symptoms at 28 dpi in the two experiments.

**Table 2 viruses-14-00499-t002:** Infectivity of infectious constructs of AYVV-NT harboring different mutations in C4 protein on *N. benthamiana*.

	Infectivity ^1^			
Inoculum	Experiment I	Experiment II	Experiment III	Leaf Curling Symptom
Buffer	0/1	0/1	0/1	-
pBin19	0/1	0/1	0/1	-
pBinAYVV	1/1	1/1	4/4	Severe upward
A-mut	4/10	9/9	4/4	Mild upward
B-mut	ND ^2^	14/14	5/5	Severe upward
C-mut	ND	ND	4/4	Severe upward

^1^ Infectivity: the number of infected plants/number of plants inoculated. ^2^ ND: not done.

**Table 3 viruses-14-00499-t003:** Effect of A-mut on leaf curling symptoms caused by pBinAYVV.

	Experiment I		Experiment II		Experiment III	
Inoculum	Infectivity ^1^	Attenuation ^2^	Infectivity	Attenuation	Infectivity	Attenuation
pBinAYVV	4/4	0/4	4/4	0/4	4/4	0/4
A-mut	4/4	4/4	4/4	4/4	3/4	3/4
pBinAYVV → A-mut	2/4	0/4	ND ^3^	ND	4/4	0/4
A-mut → pBinAYVV	3/4	3/4	3/4	3/4	3/4	3/4

^1^ Infectivity: the number of plants infected by the first construct/number of plants inoculated. ^2.^ Attenuation: the number of plants showing attenuated leaf curling symptom/number of plats inoculated. ^3^ ND: not done.

## Data Availability

The full-length sequences of AYVV-NT, TLCV-TC, and SqLCV-YL genomes have been deposited in GenBank under the accession numbers EF458639, MZ713252, and EU479710/EU479711 (for SqLCV-YL DNA A/B) respectively.

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
