# Peer review of "Identification of Crucial Amino Acids in Begomovirus C4 Proteins Involved in the Modulation of the Severity of Leaf Curling Symptoms"

_viruses, 2022, doi:10.3390/v14030499_

Round 1

Reviewer 1 Report

  - Dai et al raised an interesting question when studying the determinants of leaf curling symptoms  caused by begomoviruses. In this article, they showed  that  (i) two different begomoviruses caused leaf curling with different orientation on N. benthamiana, (ii) the severity of  the leaf curling symptom  depends on amino acids at position 8 and 14 of the C4 gene  and (iii) preinoculation of plants with a mutant inducing mild symptoms attenuates the severity of symptoms caused by the wild type virus inoculated 7 days after.

The article is easy to read and has been well improved.

My concern are the followings:

 The recombinant sequence includes the 3’part of the intergenic region that contains some determinant for gene expression and regulation. We have no idea of the nucleotidic differences between AYVV and TLCV in this region. In addition to the sentence lines 421-422 authors could show the sequence comparison between both viruses in a supplementary file (region 1-136)

To my opinion, the control with preinoculation with Pbin or inocultation on   older plants line 445 is very important and should be added in the results; It is not enough to write that it will be done later; it is an easy experiment which can be performed quickly

What about virus accumulation? Is it possible to imagine that differences of severity or leaf curling direction are related to the level of virus accumulation in plants?

Title 3.2: Even if authors wanted to identify the determinants of leaf curling direction, this is not what they found, so the title should be modified, or at least, it should be mentioned in the conclusion lines 333 that the hypothesis could not be confirmed on leaf curling direction

In figure 3, showing electrophoregrams is unnecessary. sequence alignment is better

Author Response

Thank you very much for the critical review and helpful guidance. The comments and instructions are highly appreciated.

Please see the attachment for our responses.

Reviewer 2 Report

In my opinion, the authors have done a great job improving the manuscript and I recommend it for publication after correcting some small typographical errors.

-Line 115. Three begomoviruses

-Line 383. Independent

-Line 387. C4 protein.

Author Response

Thank you very much for the critical review and helpful guidance. The comments and instructions are highly appreciated.  

As instructed, we have revised our manuscript. The revisions and responses to the concerns raised by Reviewer 2 are detailed as follows:

Point-by-point response to the Reviewers’ comments:

Reviewer 2:

In my opinion, the authors have done a great job improving the manuscript and I recommend it for publication after correcting some small typographical errors.

-Line 115. Three begomoviruses

-Line 383. Independent

-Line 387. C4 protein.

Our response:

We sincerely thank Reviewer 2 for carefully reviewing the manuscript and correcting our errors. The above errors have been corrected in the revision accordingly. The helpful instructions of Reviewer 2 are highly appreciated.

Round 2

Reviewer 1 Report

I agree with the response provided by authors

This manuscript is a resubmission of an earlier submission. The following is a list of the peer review reports and author responses from that submission.

Round 1

Reviewer 1 Report

- Dai et al raised an interesting question when studying the determinants of leaf curling symptoms  caused by begomoviruses. For this they showed  that  (i) two different begomoviruses caused leaf curling  with different orientation on N. benthamiana, (ii) the severity of  the leaf curling symptom  depends on amino acids at position 8 and 14 of the C4 gene  and (iii) preinoculation of plants with a mutant inducing mild symptoms attenuates the severity of symptoms caused by the wild type virus inoculated 7 days after. The article is easy to read but I have some concerns on the objectives and the discussion.

The objective presented in the introduction is  “dissecting the mechanism underlying the orientation of leaf curling symptoms. However, it appears that even if  in the first part of the work they found that AYVV and TLCV caused leaf curling of different orientation on N. benthamiana, the rest of the paper focuses on the level of severity of leaf curling and premunition, but not leaf curling orientation. I wonder why the authors did not just present the objective of identifying determinants of severity of symptoms, whatever the direction. I did not understand the importance of the direction in this study, all the more so when symptoms are different on the natural hosts. Related with this, there is no information in the introduction about the possible impact of the orientation of leaf curling, even if this is mentioned line 240. Line 40, reference 4 is cited as if it described the impact of leaf curling on yield while it is rather a  generic paper on begomoviruses. Mutant A B and C in TLCV genome were not  tested, then it is not possible to conclude on the role of these position for the severity of downward LC symptoms.

Although results of the section 3.3 are worth to publish some controls are lacking : what happens if AYVV is inoculated on plants after a pre inoculation with pbin 19? Or on  plants which are 7 days older? Table 3 : how do you explain the result 4/4 for attenuation in experiment II in plants inoculated with AYVV?

None of these results were discussed in the discussion section. The discussion is too general , focusing on interaction between C4 and host factors  and not enough on the results obtained. My opinion is that it should be re-written in relation with the results.

However I found the results interesting and worth for publication. I do not reject the article but propose to adjust objective and discussion and specify some lacks and errors in the tables and text.

Other remarks

There is a repetition of “C4 is the determinant of symptoms” lines 55, 66, in the introduction.

Authors should give accession numbers of the different viral clones in the virus description. The name of the different mutants should appear also in the text, not only in tables. Table 1: I do not know what is pBin TA3-4 or T5-20.

Title 3.2 : Even if authors wanted to identify the determinants of leaf curling direction, this is not what they found, so the title should not be about leaf curling direction. Line 259, I suppose that B mutant is F 64S

Table 2 :  Results on the wild type is missing;

Author Response

The valuable criticisms and helpful instructions of Reviewer 1 are highly appreciated. Please see the attachment for our responses to the comments of Reviewer 1.

Reviewer 2 Report

In this work, the mechanisms underlying the elicitation of the severe leaf curling symptoms caused by begomoviruses have been studied. Specifically, this work is focused on the role of the C4 proteins of three different begomoviruses in the determination of the specific direction of leaf curling symptoms in Nicotiana benthamiana. This work is very interesting and the experiments are properly done however, in my opinion, some experiments need a deeper explanation and the main text is not always clear. Please, see my suggestions and comments below.

  • I believe that the Materials and Methods section needs to be explained in more detail, for example, the protocol to construct the chimeric virus between AYVV-NT and TLCV-TC is not clear; what is an inverse-PCR?; specify the optical density used for Agrobacterium-mediated infection in all experiments, etc… Also, I suggest joining “Virus and plant materials” with “Inoculation assays” as “Plant material and virus inoculation” and “Plasmid construction” with “RCA” because those subsections contain duplicated information.
  • I think that you should explain in the main text the chimeric viruses constructed and used in this work (explained now in the legend of Figure 3). In addition, the chromatograms from the Sanger sequencing of Figure 3 are impossible to read. Explain the difference between TA3-4 and TA3-12 and between AT5-20 and AT5-24 in Table 1.
  • The title of Table 1 is not clear. Moreover, it is not clear if there are other symptoms of infection in plants infected with TA3-4/12 not showing upward leaf curling or in plants infected with AT5-20/24 not showing downward leaf curling. If the clone TA3-4 is not infectious at all, you should remove it from the table.
  • I am wondering if you have tried to transiently express the C4 proteins of AYVV-NT and TLCV-TC at a lower concentration of Agrobacterium to avoid the high levels of expression of C4 proteins.
  • Experiments described from lines 217 to 245 are all negative or unsuccessful results. In my opinion, these results should be moved to Discussion.
  • Is there any difference between experiments I and II of Table 3 that could explain the lack of attenuation driven by A-mut found in experiment I?
  • In my opinion, experiments using the natural host of these viruses are needed to demonstrate the A-mut attenuation of leaf curl symptoms.

Other minor corrections:

  • Line 92-93. Thus, these begomoviruses provide “a great tool” to study the underlying mechanisms of leaf developmental processes in N. benthamiana.
  • Figure 2 legend contains duplicated information in the main text (line 190-191).
  • Please, use the same virus names in Tables, Figures, and main text.

Author Response

The valuable criticisms and helpful instructions of Reviewer 2 are highly appreciated.  Please see the attachment for our point-by-point response to the comments of Reviewer 2.

Reviewer 3 Report

In this work authors try to elucidate which is the region involved in the leaf curling severity and direction by generating chimeric constructs between two begomovirus species that produce leaf curling symptoms in two different directions. By infection results they conclude that C4 must be the protein involved in this mechanism, and they also describe a two aminoacid motif placed in the aminoterminal side of the protein that seems to be determining the severity of the curling produced by the virus. Finally they try to use the virus with the aminoacids that produce attenuated leaf curling symptoms as a mild strain for cross-protection and they get promising results.

I find the work very interesting and thoughtful, although I think some of the infection experiments, especially the one with the A-mutant as a mild strain for therapeutic use, should be repeated at least one more time, as they have done with the “prevention” experiment. One only repetition with 3-5 plants doesn’t seem enough to be sure about the lack of effectivity, although the fact that almost all the viruses sequenced are Wt strains seems to support that is not appropriate for this use.

Now I will comment some aspects of the manuscript that I think authors should take into consideration:

Line 33: twinned

Line 35: reference 6 shouldn’t be cited here

Line 40: the reference 4 doesn’t say anything about the curling being the most important symptom for crop production. Maybe authors should look for another reference to support this idea. I think is important for the relevance of the work.

Line 67: Either “C4 proteins are” or “C4 protein is”

Line 72: I would take out “the”, it seems that C4 proteins are the only suppressors if you say it like that.

Line 74: I think reference 31 is not right, probably is a mistake and authors wanted to cite this other work of Chellappan (Chellappan P, Vanitharani R, Fauquet CM. MicroRNA-binding viral protein interferes with Arabidopsis development. Proc Natl Acad Sci U S A. 2005 Jul 19;102(29):10381-6. doi: 10.1073/pnas.0504439102. Epub 2005 Jul 8. PMID: 16006510; PMCID: PMC1177406.)

Line 76: nucleocytoplasmic

Line 83-86: although authors cite their previous work where these begomoviruses are described, I think they should say here if they are mono or bipartite.

Line 92: provide a way to/ allow us to study

Line 97: I think it would be better “as the experimental system”

Line 102: “taken together, these studies”

Line 109: leaf curling

Line 118: Reference for pUC119 should be included.

Line 142: given that the reference for bioEdit is of a version for Windows 98, maybe authors should include here also the link to the actual software that they have used.

Line 148: Names of bacteria and plant species should be in italics. This is a repeated mistake along the manuscript that should be checked and corrected by the authors.

Line 154: remained

Figure 1 legend: check italics for the species names. Lacking also the complete names of the viruses, this is a repeated mistake in all the figure and table legends.

Line 171: “for recognition by Rep protein”

Line 174: check the format for references.

Line 174: our initial attempt was to exchange the C4…

Line 176: were constructed as illustrated in figure 2..

Line 176: first 3 letters of the names of restriction enzymes should be written in italics, the other letter for the strain and the number are not. Again this is a repeated mistake along the manuscript that should be checked and corrected.

Line 177 and 183: the numbers of the nucleotides that define the exchanged segment should be the same.

Line 180: results

Line 181: results

Figure 2: both chimeric constructs should be depicted here, with their names clearly indicated, given that they are used after in several experiments. Also lacking complete names of the viruses.

Figure 3: difference of colour very evident between first and second line of pictures. Italics for names.

Table 1: Where I can find the information about pBin AT5-20 or pBin TA3-4? They are not cited or described in the text. And even for the other two chimeras, pBIN AT5-24 and pBin TA3-12, the description of the construct should be included in the table legend in order to follow the results.

Line 219: Have the authors tried PVX instead of this BaMV vector? It is the most usual vector for expressing viral protein in plants.

Line 222: We had to resort to ??? I recommend revise this sentence.

Line 234-236: I would put here not “suggesting” but “confirming” given that it has already been described for many monopartite and also for some bipartite geminiviruses  that C4 is essential for infectivity.

Figure 4: line 294, begomoviruses

Table 2: only one assay for C-mut mutants seems to be not enough to be sure about the lack of effect on infection.

Figure 5; pictures of complete plants seem a little dark and small.

Line 334: I don’t see really a lower infectivity for A-mut, only a lower production of symptoms. Maybe I don’t get it right, but I understand that the preventive effect of this A-mutant on the plants is because this attenuated virus is able to replicate and infect successfully the plant and as it has arrived before, when the other Wt virus is inoculated, is not able to infect the plant. Is it right?

Table 3; for the therapeutic experiment I think only one repetition is not enough to conclude the lack of effect.

Line 364: in the discussion authors talk about using a virus with a reverse direction of curling to neutralize the symptoms, but they haven’t checked this using both Wt viruses, only the mutant version A of AYVV. Have they tested it? Do they plan to do it? It would be nice to have it in the discussion.

I also would find interesting if they had done also double or triple mutant combining A, B and C mutations in AYVV to check if they obtained a change in leaf curling direction, and not only a decrease in the upward leaf curling intensity. Are they planning to do it?

Line 402: erratum “leaf”

.

Author Response

The valuable criticisms and helpful instructions of Reviewer 3 are highly appreciated.  Please see the attachment for our point-by-point response to the comments of Reviewer 3.
